# Predicting Large Model Test Losses with a Noisy Quadratic System

**Chuning Li** [1] [2]   **Chris J. Maddison** [1] [2]

## Abstract

We introduce a predictive model that estimates the pre-training loss of large models from model size ($N$), batch size ($B$) and number of weight updates ($K$). This is the first loss prediction model that can handle changing batch size. The model outperforms Chinchilla's loss model, a model of the test loss using the batch size and number of tokens, in terms of projecting the loss at extrapolated compute budgets (up to 1000 folds). A natural use of the model is to find optimal $N, B, K$ configurations under explicit and compound resource constraints like time, memory and compute. In our experiments, the model-selected configurations are close to ground-truth optimal. Our work advocates for loss prediction as a better alternative to heuristic-based laws, which are growing in complexity. The implementation is available on GitHub.

## 1. Introduction

During the late 2010s, deep learning researchers observed that the cross-entropy loss of large models (LMs) follows a power law in terms of the amount of pre-training compute (Hestness et al., 2017; Kaplan et al., 2020), launching the research area of neural scaling laws. Since then, the field has expanded to address questions of how to optimally select pre-training configurations as models scale up.

One approach, pioneered by the widely applied Chinchilla scaling study (Hoffmann et al., 2022), made use of a simple loss model that estimates test loss with model size ($N$) and dataset size ($D$), which is then used as a surrogate to LM pre-training to find the optimal $N, D$. However, as the number of pre-training variables grows, it becomes harder to find a descriptive and computable functional form for the loss model (Muennighoff et al., 2025; Luo et al., 2025).

An alternative approach is to find stable patterns directly in the *optimal* pre-training configurations. For example, fitting a power law relationship between the optimal token budget and the number of training iterations (Bergsma et al., 2025), or using special initializations so that the optimal learning rate remains constant with scale (Yang et al., 2022).

However, a "loss-model-free" approach to scaling comes with its own challenges: (1) heavy reliance on human ingenuity to discover patterns; (2) ambiguity over how multiple rules interact (e.g. Zhang et al. (2024) studied how the optimal selection of batch size $B$ interacts with that of model size $N$), and (3) laws that are a few levels removed from loss prediction are hard to evaluate rigorously.

In this paper, we introduce a loss model family that maintains the conceptual clarity of a Chinchilla-style loss model, but is more easily extended towards multiple pre-training variables on account of being mechanistic.

Specifically, we introduce a predictive loss model that expresses the test loss as a function of $N, B, K$, and we name it the Noisy Quadratic System (NQS). NQS naturally models the interaction of batch size and model size, supporting optimization under complex and compound constraints like specific time-memory-compute budgets. The NQS model class can be evaluated on a strict train/holdout split to gauge its deployment time performance.

The NQS is inspired by theoretical models of neural network (NN) scaling dynamics (Zhang et al., 2019; Maloney et al., 2022; Paquette et al., 2025; Lin et al., 2025; van Laarhoven, 2017). Like many of these models, NQS models LM test losses as the expected risk of a stochastic optimization algorithm that mimics the training of the LM.

Unlike the theoretical models, however, the NQS has a simple closed-form solution that admits numerical approximation. Such a numerical approach eliminates the need for the case-by-case mathematical analysis of traditional asymptotic theoretical models. For example, theoretical results considering mini-batch noise suggest several asymptotic phases with distinct functional forms (Paquette et al., 2025). NQS naturally adapts between these phases via inferred parameters.

Despite modeling the full trajectory of an optimization process, the NQS is as lightweight as Chinchilla. Using GPUs,

---

[1]University of Toronto [2]Vector Institute. Correspondence to: Chuning Li <chuning.li@mail.utoronto.ca>, Chris J. Maddison <cmaddis@cs.toronto.edu>.

*Proceedings of the 43rd International Conference on Machine Learning*, Seoul, South Korea. PMLR 306, 2026. Copyright 2026 by the author(s).

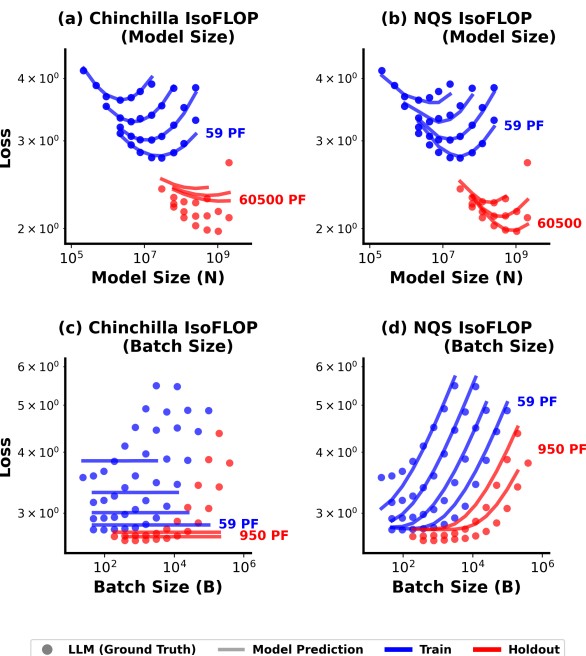

*Figure 1.* The Noisy Quadratic System (NQS) predicted LLM test losses on holdout data, outperforming Chinchilla's loss model. Shown are "IsoFLOP" slices: data points arranged by their pre-training compute budget (selectively labeled next to the curves, in PetaFLOPs). *Top*: NQS successfully predicted the loss of LLMs over variations of model size $N$, holding constant the total FLOPs budget, at compute budgets up to 1000 times higher than its training data. *Bottom*: NQS can predict variations in LLM test loss due to changes in batch size $B$, holding constant the model size $N$ and total number of tokens $D$. As far as we know, NQS is the first method that can make such predictions. We use the loss model in Chinchilla Method 3 as our baseline. The figure is based on a series of Pythia-like models of sizes up to 2B and pre-training compute budgets between 1 and 60,000 PetaFLOPs.

the wall-clock time to run the inference procedure for NQS is comparable to that of Chinchilla method 3. We plan to release a package that will make NQS as easy to use as the Chinchilla loss model.

In our experiments, the NQS loss model was also better at loss prediction than Chinchilla method 3 (Fig.1). For a fair comparison with Chinchilla, we used the exact method from Besiroglu et al. (2024), and we compared the performance on a holdout set so that the more complex NQS model class did not have an inherent advantage. We also explored other ways to improve the performance of the Chinchilla loss model on our dataset (App.2).

Our work advocates loss modeling as the better approach to scaling laws, over case-by-case heuristic analysis. Loss models allow us to make precise, quantitative hypotheses about LM pre-training dynamics. They also provide a practical approach to pre-training configuration design.

## 2. How Good Is Chinchilla as a Predictive Model of Large Model Test Loss?

Chinchilla method 3 is the quintessential loss model for LLMs. It gives an estimate of the pre-training test loss as a function of the number of parameters in the LLM ($N$) and the number of tokens used to train the LLM ($D$). The intended use case of the Chinchilla loss model is to select an $N/D$ ratio, and its loss estimation formula is used to identify a pair of $N, D$ that minimizes pre-training loss, holding constant the total compute budget $C \approx 6ND$.

Chinchilla uses the following parametric form to model LLM test losses. We re-parameterized it for consistency with our predictive model's semantics (introduced later).

---

**Chinchilla Method 3, Pre-Train Test Loss Model**

$$L_\theta^{\text{CHIN}}(N, D) = \mathcal{E}_{\text{irr}} + \frac{P}{N^{p-1}} + \frac{Q}{D^{p/q-1/q}}, \quad (1)$$

$D = B \times K \times$ seq. length, is the total number of tokens used in training, where $B$ is batch size, and $K$ is the number of batches processed, and $\theta = (p, P, q, Q, \mathcal{E}_{\text{irr}}) \in (\mathbb{R}^{\geq 0})^5$ are parameters of the loss prediction model, satisfying $p > 1$.

---

To use the formula, we must first estimate the parameters $\theta$. Chinchilla's inference procedure is as follows:

1. A series of LLMs from a particular architecture family were run, so as to obtain a dataset of the form $\mathcal{D} = \{(N_i, D_i, L_i)\}_{i=1}^m$ where $L_i$ is the test loss of a model of size $N_i$ trained on $D_i$ number of tokens.

2. Compute the average error of the predictions: $\mathcal{L}(\theta; \mathcal{D}) = \frac{1}{m} \sum_{i=1}^m \text{Huber}(\log L_\theta(N_i, D_i), \log L_i)$.

3. Find $\theta$ by minimizing $\mathcal{L}$: $\theta^* = \text{argmin}_\theta \mathcal{L}(\theta; \mathcal{D})$. This is solved by an L-BFGS algorithm. In this paper, we use an algorithm based on the original Hoffmann algorithm, but improved by Besiroglu et al. (2024).

**How well does Chinchilla perform as a predictive loss model?** For a loss model to be useful for scaling tasks, it is essential that the model can predict the loss of more compute intensive LLM runs based on information from smaller LLM runs. A good loss model should therefore be tested on its ability to perform out-of-distribution, and particularly to extrapolate across compute scales. We conducted a small experiment to understand if Chinchilla is up to this task.

We obtained the original data set from the Hoffman et al. study, extracted by Besiroglu et al. (2024), and fitted the Chinchilla model on this dataset using the procedure outlined above.

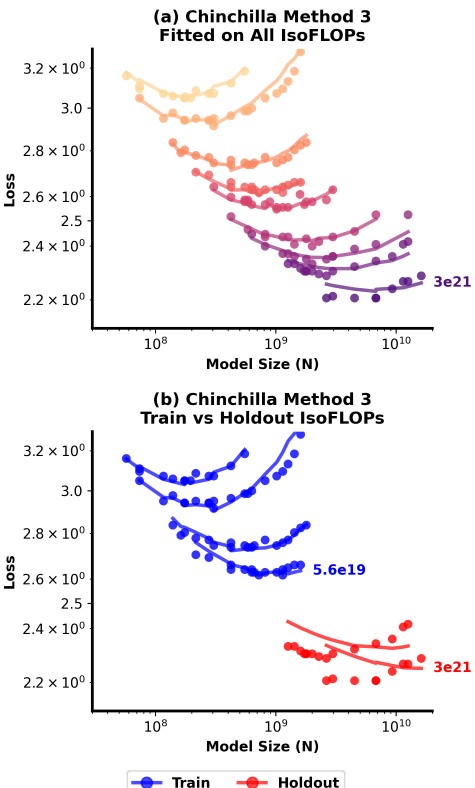

*Figure 2.* Chinchilla Method 3 is not a great predictive model. (a) Chinchilla Method 3 fitted on the entire Hoffmann IsoFLOPs dataset describes the data well. (b) Once the dataset is divided into Train/Holdout, the performance on the holdout slices deteriorated. For this figure, we used the original Hoffmann series of LLM data points upon which the Chinchilla method was developed (Besiroglu et al., 2024).

First, we tested Chinchilla's in-distribution predictive performance. We fit the model on the entire IsoFLOPs partition of the Hoffman data, and obtained a good fit of the data, including for the data points at the largest training budgets (see Fig. 2(a)).

Then, we tested Chinchilla's out-of-distribution predictive performance. We fit the model on a low-compute "Train" subset and tested it on a high-compute "Holdout", leaving a compute gap of 50× between the largest Holdout FLOP budget and the largest Train FLOP budget. Chinchillla's Holdout performance deteriorated in this experiment (see Fig. 6(b)). As the compute gap between the training and holdout dataset increased, the deficiency of Chinchilla as a loss prediction model became more dramatic (Fig.6 in App.B). As we will see, the NQS is a stronger predictive model in this sense: in our experiments it extrapolates better across compute scales.

## 3. Background on Training Dynamics

The NQS is inspired by three distinct lines of work on NN training dynamics.

### 3.1. Inspiration One: Linear Regression Dynamics

One line of work studies NN training and scaling dynamics by drawing analogies to learning in a special class of linear regression models. Similar to LLMs, as the dataset size and model dimension increase, the expected risk of these regression problems improves following a power law. For a high-level understanding, we present a model similar to Bordelon et al. (2024), which is a good representative of these works.

---

**Linear Regression Models of Training Dynamics**

*Modeling Assumptions*

- Let $X \in \mathbb{R}^M$ be a very high-dimensional latent input. The covariance matrix of $X$ is diagonal with eigenvalues $\lambda_i \propto i^{-q}$.

- Labels $y$ are generated from the latent space via a linear map, $y = (w^*)^\top X + \epsilon$, where $w^*$ is a fixed vector satisfying a power $(w_i^*)^2 \lambda_i \propto i^{-p}$ and $\epsilon$ is additive noise.

Instead of observing $X$ directly, the modeler observes a lower dimensional variable $\Phi = PX$, where $P \in \mathbb{R}^{N \times M}$ is a random Gaussian matrix mapping $X$ into a lower dimensional subspace. The modeler is given a dataset $\{(y_i, \Phi_i)\}_{i=1}^D$ from which to learn a good linear predictor of $y$ from $\Phi$.

*Expected Risk.* The expected risk $L^{\mathrm{Lin.Reg.}}$ of an SGD estimator improves as $N$ and $D$ increases.[a] The expected risk decomposes,

$$L^{\mathrm{Lin.Reg.}} = \mathcal{E}_{\mathrm{irr}} + \mathcal{E}_{\mathrm{appx}} + \mathcal{E}_{\mathrm{bias}} + \mathcal{E}_{\mathrm{var}}, \quad (2)$$

into four terms: the irreducible error $\mathcal{E}_{\mathrm{irr}}$ due to label noise $\epsilon$, the approximation error $\mathcal{E}_{\mathrm{appx}}$, due to the fitted regression model being of lower dimension than the data generating distribution, and the bias $\mathcal{E}_{\mathrm{bias}}$ and variance $\mathcal{E}_{\mathrm{var}}$ of SGD.

---
[a]Maloney et al. (2022) solved the regression problem via ERM while later work uses gradient flow (Bordelon et al., 2024) or SGD (Paquette et al., 2025).

---

We can think of the Chinchilla loss model as an asymptotic approximation of $L^{\mathrm{Lin.Reg.}}$: it can be shown that $\mathcal{E}_{appx} \sim \frac{P}{N^{p-1}}$ and $\mathcal{E}_{bias} \sim \frac{Q}{D^{p/q - 1/q}}$ (Bordelon et al., 2024); but the Chinchilla loss model does not account for the the variance term $\mathcal{E}_{\mathrm{var}}$ in $L^{\mathrm{Lin.Reg.}}$.

The variance term $\mathcal{E}_{\text{var}}$ is harder to analyse asymptotically, admitting different functional forms in different scenarios (Paquette et al., 2025). This could explain why batch size is difficult to model: mini batching adds noise to the system and magnifies the influence of the variance term $\mathcal{E}_{\text{var}}$.

The data generation and modeling assumptions have nice interpretations. The assumption that the covariance of $X$ follows a power law is consistent with the experimental finding that the spectra of LLM Hessians satisfy power laws (Tang et al., 2025). The power law in $w^*$ says that, for perfectly optimized $w$, adding dimensions to the model provides marginal improvements in the loss that diminish as more dimensions are added.

### 3.2. Inspiration Two: The Noisy Quadratic Model

The Noisy Quadratic Model (NQM, Zhang et al., 2019) is a model of NN training dynamics under rotation-invariant optimizers, based on the optimization of a simple quadratic function. In contrast to linear regression models, the NQM incorporates a gradient estimator that allows Zhang et al. to provide a precise analysis of the bias and variance terms for a variety of stochastic optimizers. As the authors show, the NQM qualitatively matches the response of NN training to changes in batch size.

---

**The Noisy Quadratic Model**

*Expected Risk.* The NQM models the training dynamics of NN test losses as the expected risk of stochastic optimization on a quadratic loss surface:

$$L^{\text{NQM}} = \mathbb{E}\left[\sum_{n=1}^{N} n^{-1}(w_n^{(k)})^2\right],$$

where (for example)

$$w_n^{(k)} = (1 - \gamma n^{-1})w_n^{(k-1)} + \gamma \xi_n^{(k)},$$

$\xi_n^{(k)} \sim \mathcal{N}(0, n^{-1}\frac{1}{B})$, and $\gamma$ represents the learning rate. Zhang et al. also study Polyak averaging and momentum in this model.

---

**Can we use the NQM to develop a predictive loss model?** The NQM model is lightweight and computable, but incomplete for loss prediction. $L^{\text{NQM}}$ only represents the estimation error (bias and variance). Naively increasing $N$ would increase the value of $L^{\text{NQM}}$, which doesn't match the dynamics of LM test losses. In the linear regression models presented above, this was corrected by $\mathcal{E}_{\text{appx}}$.

### 3.3. Inspiration Three: The Effect of LayerNorm

LayerNorm and other normalization techniques have a large effect on training dynamics (van Laarhoven, 2017; Li et al., 2020; Hoffer et al., 2019). The effect of normalization is particularly pronounced for small batch training, where the variance term is larger. Intuitively, LayerNorm restricts the size of activations, thus controls the magnitude of the gradients, and therefore has a similar effect to reducing the learning rate. van Laarhoven (2017) proposed the following as a simple model of how LayerNorm reduces the learning rate during LM training:

---

**LayerNorm and the Effective Learning Rate**

In deep learning models trained with LayerNorm, the Effective Learning Rate is $\gamma_{\text{eff}} \propto \frac{1}{||w||^2}$.

---

This effect is not automatically captured by the linear regression models or the NQM. We will see an ablation study in the next section (Fig. 3).

## 4. The Noisy Quadratic System

Building on the background from the previous section, below we go through the assumptions and derivations to give an intuition of the inner workings of the NQS. The derivation illustrates NQS's mechanistic nature, but for all practical purposes, it suffices to know the relatively simple expressions in (5) and (6).

We start by simplifying the linear regression models (inspiration 3.1), incorporating insights from the NQM (inspiration 3.2). Then we add in the effect of LayerNorm so that the NQS can model small batch training (inspiration 3.3).

### 4.1. A Lightweight Relative of the Linear Regression Model

Similar to the linear regression models, we model the test loss of LLMs as the risk of the stochastic optimization of a quadratic function, but: (1) we simplify away the random projection matrix $P$, replacing it with a fixed projection; (2) learning from the NQM, we set mini-batch noise $\propto 1/B$; (3) in addition to $p, q$, we give the noise structure its own power law spectrum, parameterized by $r$, for added flexibility. As a result, we obtain the following quadratic function $\mathcal{Q}^{\text{NQS}}$.

Let $w_m^* \in \mathbb{R}^{\mathbb{N}}$ be a square-summable sequence, $H : \mathbb{R}^{\mathbb{N}} \mapsto \mathbb{R}^{\mathbb{N}}$ a positive-definite linear mapping between sequences[1], and $\mathcal{E}_{\mathrm{irr}} \geq 0$. For $w \in \mathbb{R}^{\mathbb{N}}$, define

$$\mathcal{Q}^{\mathrm{NQS}}(w) = \mathcal{E}_{\mathrm{irr}} + \tfrac{1}{2}\langle w - w^*, Hw - Hw^* \rangle. \quad (3)$$

$\langle w, v \rangle = \sum_m w_m v_m$ is the standard inner product. $w \in \mathbb{R}^{\mathbb{N}}$ represents an LLM, $w^*$ is the best LLM achievable in our model family, $\mathcal{E}_{\mathrm{irr}}$ is the best achievable loss (the Bayes error if the model family is a universal function approximator), and $\mathcal{Q}$ is the expected test loss.

We can think of the top $N$ eigen directions of $H$ as the trainable parameters of an LM and the remaining directions as *latent*, untrained parameters; this corresponds to the linear regression model projecting the very-high-dimensional input $X$ to $N$-dimensional via the random matrix $P$ (Sec.3.1). We thus model large model training as stochastic gradient descent, traveling along the finite $N$-dimensional subspace corresponding to the highest eigen directions. Formally:

Let $v_n$ be an orthonormal basis of $H$'s eigenvectors, in non-increasing order of the eigenvalues $\lambda_n$. Let $\gamma, R > 0$, $w^{(0)} \in \mathbb{R}^{\mathbb{N}}, \xi_n^{(k)} \in \mathbb{R}$ be random, and $\mathbb{W}_N = \mathrm{span}\{v_n\}_{n=1}^N$ for $N > 0$. Define the update: $w^{(k)} =$

$$w^{(k-1)} - \gamma \, \mathrm{Proj}_{\mathbb{W}_N}\left(Hw^{(k-1)} - Hw^*\right) + \gamma \sum_{n=1}^N \xi_n^{(k)} v_n. \quad (4)$$

This is an SGD optimizer of $\mathcal{Q}$ that updates $w$ along the top $N$ eigen directions of $H$ with noise injected along the same subspace.

Analogously to those from the theoretical models of NN training dynamics, we make the following assumptions:

*Assumption 4.1* $\mathbb{E}[\lambda_n\left(\langle v_n, w^{(0)} - w^* \rangle\right)^2] = \frac{P}{n^p}$,

*Assumption 4.2* $\lambda_n = \frac{Q}{n^q}$,

*Assumption 4.3* and $\xi_n^{(k)} \sim \mathcal{N}\left(0, \frac{R}{n^r}\frac{1}{B}\right)$ indep.

Assumptions 1 and 2 are analogous to those of the linear regression family of models. The values $p, q$ correspond to those in the Data Generation assumption described in 3.1.

Assumption 3 is adapted from the NQM's assumption on $\xi$. The NQM assumed that $\lambda_n$ and $\xi_n$ share the same power-law exponent ($q = 1$). In our case, we allow the exponent of $\xi_n$ to move freely with the parameter $r$.

The *Noisy Quadratic System* is the set of all functions that can be described as the expected value of $\mathcal{Q}$ after $K$ steps of update (4). The common expression for these functions is straightforward, as follows: [2]

---

[1]Technically, we also assume that $H$ is compact and self-adjoint, to invoke the spectral theorem.

[2]The NQS model class has at most 7 degrees of freedom; the expected value of $\mathcal{Q}$ is invariant to changes in the eigenbasis of $H$ and the step size $\gamma$ is redundant. We prove this in the appendix.

**Definition 4.1. (NQS Model Class)** For integers $N, B, K > 0$, the *Noisy Quadratic System* model class consists of functions satisfying $L_\theta^{\mathrm{NQS}}(N, B, K) =$

$$\mathcal{E}_{\mathrm{irr}} + \underbrace{\sum_{n=N+1}^{\infty} \frac{P}{n^p}}_{\mathcal{E}_{\mathrm{app}}(N)} + \underbrace{\sum_{n=1}^{N} \frac{P}{n^p}\left(1 - \frac{\gamma Q}{n^q}\right)^{2K}}_{\mathcal{E}_{\mathrm{bias}}(N,K)}$$

$$+ \underbrace{\sum_{n=1}^{N}\sum_{k=1}^{K} \frac{\gamma^2 QR}{Bn^{q+r}}\left(1 - \frac{\gamma Q}{n^q}\right)^{2K-2k}}_{\mathcal{E}_{\mathrm{var}}(N,B,K)}, \quad (5)$$

where $p > 1, P, q, Q, R, r, \gamma > 0, \mathcal{E}_{\mathrm{irr}} \in \mathbb{R}$ and $\theta = (p, P, q, Q, r, R, \mathcal{E}_{\mathrm{irr}})^a$ are the learnable parameters.

---

[a]The learning rate parameter $\gamma$ is redundant given $P, Q$ (see App.E.1).

## 4.2. Modeling the Effect of LayerNorm

Applying inspiration 3.3, $\gamma_{\mathrm{eff}} \propto \frac{1}{||w||^2}$, to NQS requires us to dynamically update the learning rate $\gamma$ given the current weight norm ($||w_k||^2$), so that $\gamma_k \propto 1/||w_k||^2$.

It is straightforward to adapt Eq.5 to admit a schedule of learning rates that updates its value based on the current state of the system. We provide the expression in App.E.2.

To compute $\gamma_k \propto 1/||w_k||^2$, we need to know the relative position between the initial weights $w_0$ and the origin. We add a parameter, in addition to $\theta$, that represents the weight norm at initialization, $s = \mathbb{E}[||w^{(0)}||^2]$, and assume that:

*Assumption 4.4*: the initialization $w^{(0)}$ is independent of the other random quantities in the system.

We then approximate $||w^{(k)}||^2$ with its expected value $\mathbb{E}[||w^{(k)}||^2]$ using $s$.

**Modeling The Effect of LayerNorm**
Periodically update the weight update rule (4) to reset the learning rate to:

$$\gamma_k = \gamma_{\mathrm{init}} \times \frac{\mathbb{E}[||w^{(0)}||^2]}{\mathbb{E}[||w^{(k)}||^2]}, \quad (6)$$

where we parametrize $\mathbb{E}[||w^{(0)}||^2] = s > 0$ and $\mathbb{E}[||w^{(k)}||^2]$ can be expressed as a function of $\theta, s, N, B$ and $K^a$.

---

[a]See App.E.2 for the full expression.

The ablation study in Fig.3 showed that the vanilla NQS (5) is already a good predictive model for large batch training,

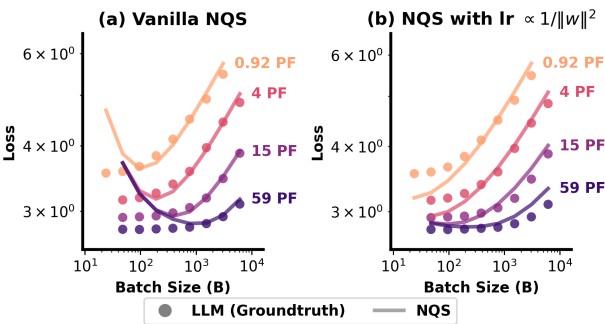

*Figure 3.* Accounting for the effect of LayerNorm by setting the learning rate $\gamma \propto \frac{1}{||w||^2}$ helps NQS fit large models trained with small batch sizes.

but adjusting for the effect of LayerNorm is necessary for NQS to perform well on smaller batch sizes.

From this point on, to experiment with the NQS, it is sufficient to rely on the simplified expressions of (5) and (6).

### 4.3. Inferring NQS parameters

The procedure to find $\theta$ for NQS is very similar to that of Chinchilla's presented in Sec.2. For predicting models trained at large batch sizes, step 4 below is optional.

1. A series of large models from a particular architecture family were run, so as to obtain a dataset of the form $\mathcal{D} = \{(N_i, B_i, K_i, l_i)\}_{i=1}^{m}$ where $l_i$ is the test loss of a model of size $N_i$ trained with mini-batch size $B_i$ for $K_i$ number of weight updates.

2. Define the error of the model as: $\mathcal{L}_\theta(\mathcal{D}^{\text{Train}}) = \frac{1}{m} \sum_{i=1}^{m} \left( \log L_\theta(N_i, B_i, K_i) - \log l_i \right)^2$.[3]

3. Find $\theta$ that minimizes $\mathcal{L}$: $\theta^* = \operatorname{argmin}_\theta \mathcal{L}(\mathcal{D}^{\text{Train}})$. An exact solution is intractable. We use a gradient-based method [4] to travel down the loss surface, parallelizing over multiple initializations for efficiency.

4. The above procedure is run on $B_i$ sufficiently large, so that the LayerNorm adjustment is not strictly necessary. Then we examine a set of runs with smaller $B_i$'s to guide the selection of $s$.

   A natural starting point for $s$ is an estimate of the expected weight norm of the large model at initialization. In our main experiment, we set $s = N \times 0.02^2$, where 0.02 is a common standard deviation for LLM weight initialization. In the general case, we recommend performing a grid search over plausible values of $s$, and selecting an $s$ that minimizes the error over the small-

batch data set $s^* = \operatorname{argmin}_s \mathcal{L}(\theta^{*,\mathcal{D}_{\text{largeB}}}, s, \mathcal{D}_{\text{smallB}})$, which can be computed efficiently.

### 4.4. NQS is Fast to Compute and Train

Luckily, equation (5) can be computed efficiently. To address the dependence on $N$, we estimate the sum over $N$ with a numerical integral over $N$, using the Euler-Maclaurin formulae (Apostol, 1999). To address the dependency on $K$, we use the geometric series summation formula . The final computational cost is $\mathcal{O}(1)$ in $N, K$.

To infer $\theta$, it is necessary to compute the gradient of Eq. (5) with respect to $\theta$. The gradient is computed analogously, by integrating over $N$ the gradient of the summand w.r.t. $\theta$.

Evaluations of (5) take less than 1 second on our hardware and fitting $\theta$ takes less than 5 minutes. In our experiments, with parallelization, NQS inference runs faster than Chinchilla Method 3.

## 5. Experiments

### 5.1. Loss Prediction

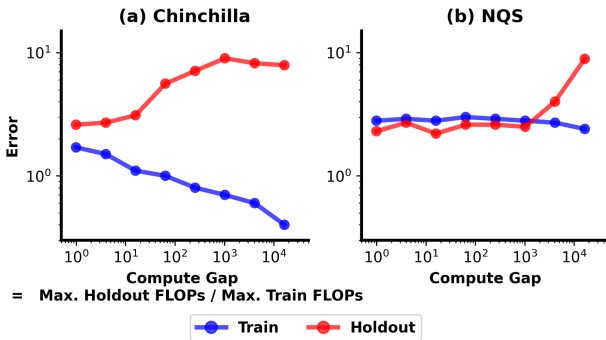

*Figure 4.* NQS is robust in extrapolation. The x-axis represents gradually removing data points from the training set, starting from those training points with the highest compute budgets, so as to increase the compute gap between the training set and the holdout set. (a) In our experiments, Chinchilla can successfully extrapolate 20 times into higher compute budget. (b) NQS succeeded until the largest training run is reduced to 1/4000 of the largest test run in compute (beyond which point only two IsoFLOPs slices remain in the training set). NQS train loss and test loss stayed close until that point. Error is the average Huber loss, in units of $10^{-5}$, evaluated over the IsoFLOPs holdout data.

***LLMs.*** We prepared two sets of LLM runs to assess the loss prediction performance of the NQS.

In the main experiment, we trained a granular (across model sizes) version of the Pythia model family[5] with model sizes up to 2B and compute budgets between 9e14 and 6e19 FLOPs. We trained LLMs on OpenWebText2 (Gokaslan

---

[3]Using the Huber loss gives similar results.
[4]Please see Appendix D.2 for details.

[5]Please see the appendix for definitions of the model family

*Table 1.* NQS predicts LLM loss on held-out data, outperforming Chinchilla. The metric shown is the average Huber loss, denoted in multiples of $10^{-5}$. The average Huber loss is defined as $\mathcal{L} = \frac{1}{m} \sum_{i=1}^{m} \text{Huber}(\log L^{\text{model}}(N_i, B_i, K_i),\ \log l_i)$. We use the Huber loss to be consistent with the Chinchilla training objective. For an evaluation with the mean squared error, please refer to App.C

| **Pythia + OWT2**, up to ×1024 Compute Gap | | | |
|---|---|---|---|
| | Train | Holdout | |
| Method | | IsoFLOPs | B-K Plane |
| Chinchilla | **0.8** | 9.0 | 9.8 |
| NQS | 2.8 | **2.5** | **5.6** |

| **Pythia + OWT2**, up to ×64 Compute Gap | | | |
|---|---|---|---|
| | Train | Holdout | |
| Method | | IsoFLOPs | B-K Plane |
| Chinchilla | **1.0** | 5.6 | NA* |
| NQS | 3.0 | **2.6** | NA* |

| **Llama + LM1B**, up to ×6 Compute Gap | | | |
|---|---|---|---|
| | Train | Holdout | |
| Method | | IsoFLOPs | B-K Plane |
| Chinchilla | **0.7** | 3.7 | 8.7 |
| NQS | 2.5 | **2.9** | **8.2** |

[1] *Compute Gap* $= C_{\max}^{\mathcal{D}_{\text{Train}}} / C_{\max}^{\mathcal{D}_{\text{Holdout}}}$, where $C_{\max}^{\mathcal{D}} = \max\{C_i = 6 \times N_i B_i K_i \times Seq.len : \exists l_i, s.t. (N_i, B_i, K_i, l_i) \in \mathcal{D}\}$

[2] Holdout compute budget: OWT2: 4000-60000PF for IsoFLOPs; 240-940PF for B-K Plane. LM1B: 240-940PF for both.

[*] At x64 Compute Gap, the FLOP scale of the largest training data points is at 60000PF/64, matching the maximum FLOP scale of the B-K Plane, thus a strict B-K holdout is no longer available.

& Cohen, 2019), using a customized BPE tokenizer (Gage, 1994) with a vocabulary size of 3000 and sequence length 128. Following standard practice, we train with an Adam optimizer and a cosine learning rate schedule with 1% warmup, with an initial learning rate of 1e-3.

To show that NQS can generalize over workloads, we also included results on a second, smaller set of LLMs, a family of Llama models (Touvron et al., 2023) trained on the LM1B dataset (Chelba et al., 2014). For LM1B, the maximum compute gap is x6, due to the limited number of tokens in the LM1B dataset.

**Training/Holdout Split.** We maintain a compute gap between $\mathcal{D}^{\text{Train}}$ and $\mathcal{D}^{\text{Test}}$. The compute budget for the largest

LLM in the training set is 6e16, and the compute budgets for the LLMs in the holdout set are between 2e17 and 6e19. $(\theta, s)$ of the NQS are chosen based on $\mathcal{D}^{\text{Train}}$ only.

**Details on Training Data** $\mathcal{D}^{\text{Train}}$**.** For Chinchilla, we used IsoFLOPs data, similar to those of Hoffmann et al. (2022). This dataset consisted of several compute levels, each contained a series of $N, D$ values satisfying $6ND = C$. $B$ is not specified. To obtain the most relevant data for Chinchilla, we set $B = CBS$ (Critical Batch Size), the batch size recommended by practitioner for compute-time efficiency, as originally proposed in (McCandlish et al., 2018) and recently refined by (Bergsma et al., 2025).

For NQS, we used the same IsoFLOPs data, but also added a dataset that varies in batch size. These additional data points were also arranged by compute levels, similar to those of the IsoFLOPs data, but only at two model sizes (10 mil and 100 mil): for both model sizes, a series of $B$'s surrounding the CBS, so as to model variation in batch size.

To maximize Chinchilla's performance, we did not use $B/K$ variation data in Chinchilla's training set. Because Chinchilla does not distinguish between batch sizes (given a fixed $D$), mixing CBS and non-CBS batch sizes hurts its ability to make an accurate prediction (see row 3 of Tab.2 in App.B).

**Predictive Performance.** In Tab.1, we demonstrate the performance of NQS in loss prediction over two test sets, one covering variations across model sizes, and the other across batch sizes, both at extrapolated compute scales; NQS successfully generalized to higher compute budgets in both settings and outperformed the Chinchilla loss model. We visualize the results on Pythia + OWT2 in Fig.1, and on Llama + LM1B. in App.C Fig.7.

The trends in Fig.4 shows that NQS is robust in extrapolations. We found that at up to x20 folds of extrapolation, Chinchilla's performance is comparable to NQS, but this deteriorated quickly at x100 folds and beyond. Such a trend is observed in our data as well as in Hoffmann's, the data where Chinchilla was developed on (see App. B).

The NQS is also robust to changes in the training dataset. In Fig.8 & 9 (App.C), we visualize a 90% confidence interval around the NQS predictions, from 100 trials, each trial using a random half of the training runs for inference. The confidence intervals are tight relative to the range of the predictions.

**On Complexity.** Overall, the NQS has at most 7 plus 1 degrees of freedom (see Pf.E.1 in App.E) while the Chinchilla loss model has 5. NQS is a more complex model class, and we want to make sure the comparison against Chinchilla is fair. In Tab.1, we saw that the NQS, despite its larger error

than Chinchilla on the training data, outperformed Chinchilla on holdout data. NQS's errors on Train and Holdout are comparable. This rules out NQS overfitting on Train due to its additional complexity. Within the confinement of our experiments, the additional complexity of the NQS is justified.

## 5.2. Compound Resource Allocation

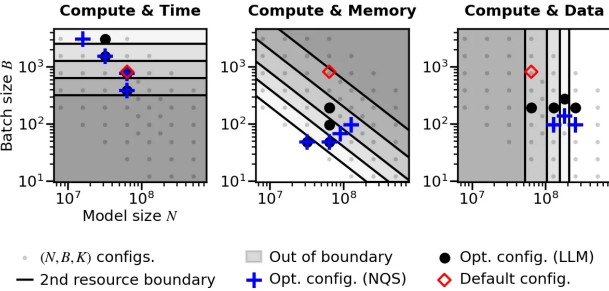

*Figure 5.* The predictions of NQS can be used to select $(N, B, K)$ under compound resource constraints. Each subplot: an IsoFLOP plane ($C = 236$ PF) with coordinates $(x, y)$ representing $N = x, B = y, K = 2.6 \times 10^{14}/(xy)$. The NQS model is trained on data with FLOP budget up to $C = 147$ PF. The red diamond is the default configuration, Chinchilla compute optimal model size trained at the Critical Batch Size (McCandlish et al., 2018). Regions satisfying progressively stricter constraints on a $2^{\text{nd}}$ resource are shaded (darker is stricter).

Using NQS to select the optimal $N, B, K$ under compound and complex constraints is straightforward. Simply compute the NQS on a grid of $N, B, K$ within the constrained set, and select the point with the lowest NQS predicted loss. An example of such compound constraints is a compute-time constraint of $(c, t)$, which defines the constrained set $S_{c,t} = \{(N, B, K) :$

$K \le t, C = 6NBK \times \text{seq.len} \le c\}$, and

the NQS optimal configuration $(N, B, K)^*(c, t) =$

$$\text{argmin}_{(N,B,K) \in S_{(c,t)}} L_{\theta,s}^{\text{NQS}}(N, B, K).$$

In Fig.5, we assessed the performance of NQS under compound constraints. Namely, we set a constraint over compute ($C = 6 \times NBK \times \text{seq.len}$) and an additional constraint over one of the resources from below:

- $T = NK$ as a notion of time, where larger models are slower to train (Bergsma et al., 2025).

- $M = BN$ as a notion of GPU memory, where large models trained with large batch sizes consume more GPU memory.

- $D = BK \times \text{seq.len}$ as a measure of dataset size (in the single-epoch setting).

In all three settings, NQS consistently favored configurations that were nearly ground truth optimal.

Time and memory can be computed more precisely from the architecture, training procedure, hardware and $(N, B, K)$, but these crude approximations are sufficient to demonstrate the generality of the types of constraints the NQS can work with.

## 6. Discussions

We introduced NQS, a mechanistic loss model that is predictive of LLM test loss across variations in $(N, B, K)$. In our experiments, we found that NQS outperformed Chinchilla's loss model. We believe that NQS-like loss models can be built upon to become more powerful.

First, an improvement in the fitting algorithm could improve the fit of the NQS. In the current inference procedure, the gradient-based optimization step does not incorporate the LayerNorm adjustment, and thus $s$ is selected in a separate step (4.3). In principle, the gradient of $s$ is computable and can be optimized simultaneously with $\theta$, which will improve the fit and streamline the fitting procedure. We ran into some numerical issues in our attempt to evaluate the gradient w.r.t. to $s$, but we believe this can be addressed with careful engineering.

In our initial experiments, changes in the initial learning rate $\gamma_0$ has a greater impact on the NQS prediction than on that of the LLMs. Loosely speaking, it appears as if $|\partial L^{\text{NQS}}(N_i, B_i, K_i)/\partial \gamma_0| > |\partial l_i/\partial \gamma_0|$. We leave the following questions open, as they require more precise modeling of $\gamma_0$. (1) Can the NQS system predict the interaction of batch size and learning rates? (2) Can the NQS system optimally scale learning rate with model size and training horizons? Given the rich literature of theoretical quadratic models, we are confident that these questions would be resolved in future works.

Another promising direction for NQS-like loss models is to model higher-dimensional, or non-numerical pre-training configurations. The rolled-out optimization approach of the NQS makes it easy to extend. In the system, it is possible to use schedules (e.g. learning rates, batch size) similar to how the LayerNorm adjustment was applied. It may even be feasible to use NQS as a sandbox for developing task-specific optimizers.

## Acknowledgements

We would like to thank Roger Grosse, Tatsunori Hashimoto, James Martens, Vardan Papyan, and Percy Liang for their valuable feedback, insightful discussions, and suggestions that helped improve this work. We also thank the anonymous reviewers for their constructive comments. Resources used in preparing this research were provided, in part, by the Province of Ontario, the Government of Canada through CIFAR, and companies sponsoring the Vector Institute. We acknowledge the support of the Natural Sciences and Engineering Research Council of Canada (NSERC), RGPIN-2021-03445.

## Impact Statement

This paper presents work whose goal is to advance the field of machine learning. There are many potential societal consequences of our work, none of which we feel must be specifically highlighted here.

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

# A. Extended Related Work

**Theoretical Results on the Variance Term of the Linear Regression Models** Lin et al. (2025) assumed an exponentially decaying learning rate schedule, which results in a negligible variance term in the asymptotics, not suitable for analysing the effect of batch size. Paquette et al. (2025) discussed case-by-case the implication of noise under constant learning rate schedule and varying degrees of noise, with the noise becoming a bottleneck in late-stage training.

**Non-linear Theoretical Models of Scaling Dynamics.** More recently, theoretical models like two-layer mlps are analyzed, and found to qualitatively describe the training of NN like RNNs applied on image data (Bordelon et al., 2025; Ren et al., 2025; Arous et al., 2025). Although some of these works offer testable hypothesis (Bordelon et al., 2025), the results are limited to conjectures on the scaling exponents, and the connection with empirical results is not strong enough to warrant practical use. LLMs are underexplored in this literature.

**Practical Use of SGD Dynamics** Qiu et al. (2025) used a similar model of SGD noise dynamics that predicts loss curves, where it was observed that well-configured LLM runs exhibit typical scaling curves under given learning rate schedules (subject to normalization) that do not conform to strict power laws.

**The Search for an Optimal Batch Size.** The pressing need to utilize the parallel computing structure initiated a line of investigation to find the best batch size that balances time efficiency and compute efficiency (Shallue et al., 2019). The Noisy Quadratic Model (NQM) (Zhang et al., 2019) was found to produce useful qualitative insights in the relationship between optimizer properties and the critical batch size. Inspired by similar quadratic models, quantitative scaling laws in the critical batch size are discovered (McCandlish et al., 2018), (Zhang et al., 2024),(Bergsma et al., 2025). The idea of "gradient noise scale" (McCandlish et al., 2018) is applied in the training of large scale LLMs (Brown et al., 2020). In case where the time constraint is not severe, (Marek et al., 2025) found that smaller batch sizes are beneficial to minimizing cross-entropy under a fixed token budget; this is achievable with carefully tuned hyperparameters including those relate to the Adam optimizer.

**Scaling Laws of Learning Rate and Weight Decay.** The tuning of learning rates and weight decay are not modelled by the current version of NQS, but they are a key branch of scaling laws, and empirically influences the choice of batch size (Bi et al., 2024; Bjorck et al., 2025; Bergsma et al., 2025). For $lr$ selection, an alternative to scaling law is "hyperparameter transfer". Yang et al. (2022) prescribed a formula to configure neural networks, so that the optimal hyperparameters at a small scale also applied at a larger scale. Theoretical and empirical works followed to interpret and expand this regime (Dey et al., 2025; Everett et al., 2024).

**Scaling Models of Data.** Using the available data efficiently is key to scaling. NQS considered online training with homogeneous data, similar to (Hoffmann et al., 2022; Kaplan et al., 2020), while other works in this area explored data mixing (Shukor et al., 2025; Meta AI, 2024; Thudi et al., 2025); and training with multiple epochs (Muennighoff et al., 2025). When compared to existing practical scaling models, the NQS in its current state does not explicitly model multi-epoch training (Muennighoff et al., 2025) or data mixtures (Shukor et al., 2025), but given its close connection to theoretical works, we hope this framework can be expanded to model these configuration options and more.

**The Scaling Properties of Optimizers.** In NQS, we found that the optimization of a quadratic model with SGD, given the correct scaling parameters and proper elaborations, are practically sufficient to model NN trained with Adam (Kingma & Ba, 2017). Other works explicitly consider the scaling behavior of different optimizers (Zhang et al., 2019; Marek et al., 2025). Certain families of optimizers are found to outperform SGD in theory and in practice (Ferbach et al., 2025).

**Advocacy for Testing Extrapolation Performance of Scaling Laws.** We are not the first to advocate for testing scaling laws for its ability to extrapolate beyond seen data. Examples where laws were tested on heldout data include (Alabdulmohsin et al., 2022),Gadre et al. (2024), and more recently (Shukor et al., 2025). These works are not directly comparable to ours, as they focus on loss vs compute, over-training and data mixing respectively.

**Loss Prediction.** The idea of predicting the loss of a machine learning model long predates the advancement of language models. An early example is (Cortes et al., 1993), where the loss curves of boolean classifiers were studied.

# B. More Results on Chinchilla Method 3

*Table 2.* We chose a combination of data and optimization method that maximized Chinchilla's performance on our dataset. The metric shown is the average Huber loss, denoted in multiples of $10^{-5}$. The average Huber loss is defined as $\mathcal{L} = \frac{1}{m} \sum_{i=1}^{m} \text{Huber}(\log L^{\text{model}}(N_i, B_i, K_i), \log l_i)$, with Huber $\delta = 10^{-3}$.

| | | Train | Holdout | |
|---|---|---|---|---|
| Opt. Method | Train Data | | IsoFLOPs | B-K Plane |
| L-BFGS * | IsoFLOPs * | **0.8** | **9.0** | **9.8** |
| TRF | IsoFLOPs | 0.9 | 10.6 | 9.9 |
| L-BFGS | IsoFLOPs + B/K Variation | 1.1 | 11.8 | **9.8** |

Chinchilla Loss Model on Pythia + OWT2, up to ×1024 Compute Gap

* Used for exhibits in the main body.
[1] Opt. Methods: L-BFGS: we used Besiroglu et al. (2024)'s method with the Huber loss; TRF: we used the curve_fit function of the Scipy package (Virtanen et al., 2020) with mean squared error. Some recent papers on Chinchilla-like scaling laws replaced the L-BFGS algorithm with the Scipy package. Both methods were initialized on a grid provided by Hoffmann et al. (2022). (Gadre et al., 2024).
[2] Heldout compute budget: 4000-60000PF for IsoFLOPs; 240-940PF for IsoToken.

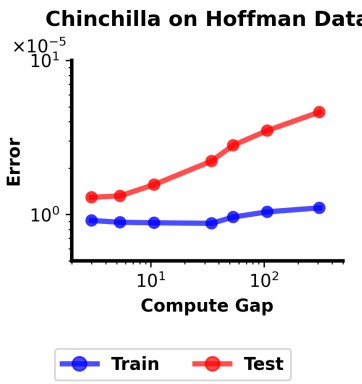

*Figure 6.* Chinchilla's extrapolation performance on the original Hoffman dataset. The Hoffman dataset spans a smaller range than our Pythia + OWT2 dataset, and was less suitable for analysis of extrapolation performance. Nevertheless, the emerging trend is consistent with results on our LLM dataset.

# C. More Results on NQS

## C.1. Performance on Llama + LM1B

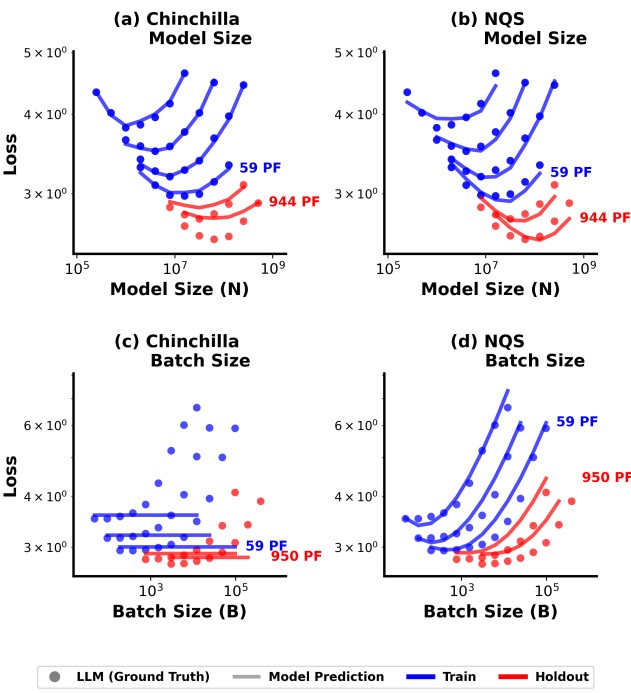

*Figure 7.* NQS performance on Llama + LM1B. The figure is based on a series of Llama-like models of sizes up to 0.5B and pre-training compute budgets between 5 PetaFLOPs and 1,000 PetaFLOPs.

## C.2. Prediction Confidence Intervals

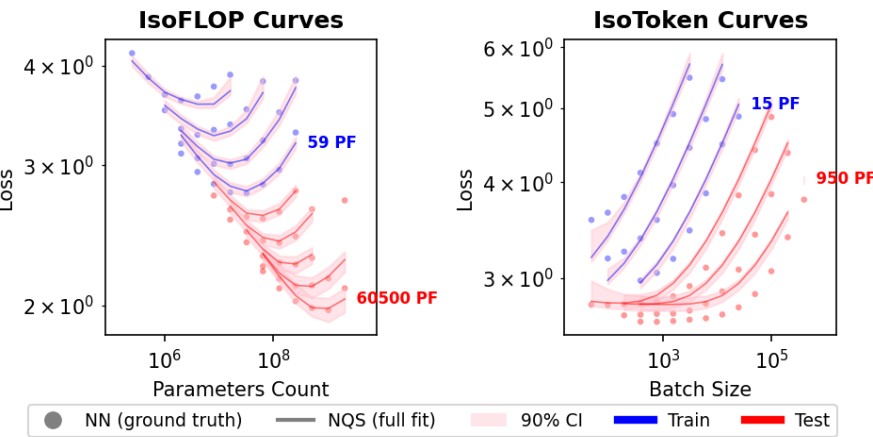

*Figure 8.* The NQS is robust to changes in the training set. Shown is a 90% confidence interval around the NQS predictions, constructed using 100 trials, each using a random subset (a 50% subsample) of the training runs to infer the NQS parameters.

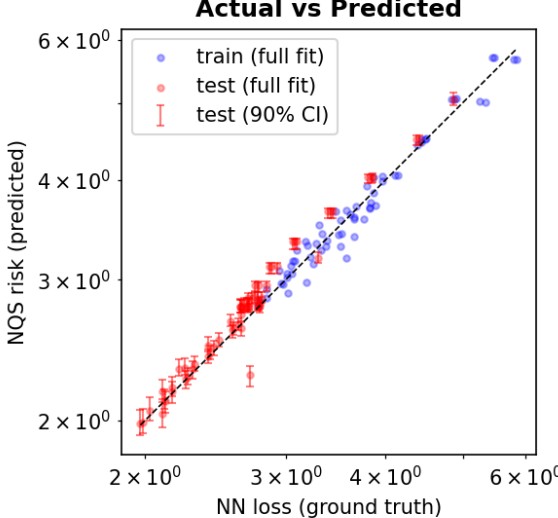

*Figure 9.* The NQS is robust to changes in the training set: the 90% confidence intervals are tight relative to the range of the predictions. The confidence intervals are constructed using 100 trials, each using a random subset (a 50% subsample) of the training runs to infer the NQS parameters.

## C.3. Main Results Measured by the Mean Squared Error

*Table 3.* NQS predicts LLM loss on held-out data, outperforming Chinchilla, as measured by the average mean squared error of the log loss, $\mathcal{L} = \frac{1}{m} \sum_{i=1}^{m} (\log L^{\mathrm{model}}(N_i, B_i, K_i) - \log l_i)^2$, denoted in units of $10^{-3}$. To be consistent with the Chinchilla training objective, in Sec. 5.1 we evaluated the loss models using the Huber loss. The Huber loss is generous about large errors, and understates the performance of the NQS on the B-K Plane. Below we provide the same table, but evaluated with the MSE.

| **Pythia + OWT2**, up to $\times 1024$ Compute Gap | | | |
|---|---|---|---|
| | Train | Holdout | |
| Method | | IsoFLOPs | B-K Plane |
| Chinchilla | **0.2** | 9.1 | 27.4 |
| NQS | 1.4 | **2.2** | **3.5** |

| **Pythia + OWT2**, up to $\times 64$ Compute Gap | | | |
|---|---|---|---|
| | Train | Holdout | |
| Method | | IsoFLOPs | B-K Plane |
| Chinchilla | **0.3** | 4.6 | NA |
| NQS | 1.5 | **2.4** | NA |

| **Llama + LM1B**, up to $\times 6$ Compute Gap | | | |
|---|---|---|---|
| | Train | Holdout | |
| Method | | IsoFLOPs | B-K Plane |
| Chinchilla | **0.0** | 2.0 | 18.8 |
| NQS | 1.2 | **1.1** | **8.1** |

[1] *Compute Gap* = $C_{\max}^{\mathcal{D}_{\mathrm{Train}}}/C_{\max}^{\mathcal{D}_{\mathrm{Holdout}}}$, where $C_{\max}^{\mathcal{D}} = \max\{C_i = 6 \times N_i B_i K_i \times Seq.len : \exists l_i, s.t. (N_i, B_i, K_i, l_i) \in \mathcal{D}\}$
[2] Heldout compute budget: OWT2: 4000-60000PF for IsoFLOPs; 240-940PF for IsoToken. LM1B: 240-940PF for both.

## C.4. Qualitative Results

The NQS model is consistent with some very recent findings on the selection of batch sizes:

- Bergsma et al. (2025) found a power law relationship between optimal batch size and token count (as opposed to between optimal batch size and total compute).

- Marek et al. (2025) showed that under a generous time budget, smaller batch sizes are preferable for higher performance.

- Zhang et al. (2024), "Empirical Takeaways" 1 to 3, namely:

    1. In Chinchilla settings, the Critical Batch Size (CBS) increases when model size $N$ and data size $D$ are jointly scaled up.
    2. If we scale up training duration $D$ while keeping $N$ fixed, the CBS increases to a similar degree.
    3. The CBS remains nearly invariant when scaling up $N$ while keeping $D$ fixed, suggesting that CBS weakly depends on model size $N$ but more strongly depends on data size $D$.

## C.5. Fitted NQS Parameters

| Parameter | SGD + Constant LR | Adam + Cosine LR |
|---|---|---|
| $p$ | 1.14 | 1.12 |
| $q$ | 0.70 | 0.59 |
| $P$ | 4.9 | 3.6 |
| $Q$ | 0.69 | 0.93 |
| $e_{irr}$ | 1.12 | 0.45 |
| $R$ | 4.5 | 4.3 |
| $r$ | 2.3 | 1.5 |

*Table 4.* Fitted NQS parameters under two sets of LLM optimizer/schedule configurations, both trained with Pythia + OWT2.

# D. NQS Algorithms

## D.1. Computation of NQS and its Gradient

This section gives details on how we efficiently compute the NQS expression (equation (5)) and its gradient with respect to the scaling parameters.

Given $(N, B, K)$ and $\theta = (P, p, Q, q, R, r, \mathcal{E}_{\text{irr}})$, the expression we would like to evaluate is

$$L_\theta^{\text{NQS}}(N, B, K) = \mathcal{E}_{\text{irr}} + \underbrace{\sum_{n=N+1}^{\infty} \frac{P}{n^p}}_{\mathcal{E}_{\text{app}}(N)} + \underbrace{\sum_{n=1}^{N} \frac{P}{n^p}\left(1 - \frac{Q}{n^q}\right)^{2K}}_{\mathcal{E}_{\text{bias}}(N,K)} + \underbrace{\sum_{n=1}^{N}\sum_{k=1}^{K} \frac{RQ}{Bn^{r+q}}\left(1 - \frac{Q}{n^q}\right)^{2K-2k}}_{\mathcal{E}_{\text{var}}(N,K,B)} \tag{7}$$

$\mathcal{E}_{\text{app}}(N)$ is computed using a JAX (Bradbury et al., 2018) implementation of the Riemann zeta function (in $\mathcal{O}(1)$ time).

For $\mathcal{E}_{\text{bias}}(N, K)$ and $\mathcal{E}_{\text{var}}(N, K, B)$:

To efficiently compute the sum of products over $K$ terms, we use the geometric sum formula, $\sum_{k=0}^{K-1} x^k = \frac{1-x^K}{1-x}$.

To efficiently compute the sums over $N$, we compute the first $5\%$ of the summation terms exactly, up till at most $N = 100$, and for the rest of the summation we approximate the sum using the corresponding integral. The integral to sum approximation is corrected with first order terms from the Euler-Maclaurin (E-M) formula. i.e. Let $L =: \min(\text{int}(0.05N), 100)$, and we evaluate an expression $\sum_{n=1}^{N} f(n)$ by

$$\sum_{n=1}^{N} f(n) = \sum_{n=1}^{L} f(n) + \sum_{n=L+1}^{N} f(n) \tag{8}$$

and

$$\sum_{n=L+1}^{N} f(n) \overset{\text{E-M}}{\approx} \int_{n=L}^{N} f(n) + \frac{1}{2}(f(N) - f(L)) \tag{9}$$

Integrals are then computed with fixed 20-point Gauss-Legendre. The run time is constant in $N$.

We explicitly calculate the first few terms in the summation, because in our experiment, these terms cannot be adequately approximated with a first-order E-M formula.

To efficiently compute the gradient $\nabla_\theta L_\theta^{\text{NQS}}(N, B, K)$, we first compute the gradient of the $N$-summands i.e., for $\nabla_\theta \sum_{n=L}^{N} f(n)$, we compute $\sum_{n=L}^{N} \nabla_\theta f(n)$. Since we implemented the computation of $f(n)$ in JAX, $\nabla_\theta f(n)$ can be implemented via $\text{jax.grad}(f)$. For the summation over $N$, analogously, we evaluate the first few terms exactly, and then approximate the rest with an integral.

$$\sum_{n=1}^{N} \nabla_\theta f(n) \approx \sum_{n=1}^{L} \nabla_\theta f(n) + \int_{n=L}^{N} \nabla_\theta f(n) + \frac{1}{2}(\nabla_\theta f(N) - \nabla_\theta f(L)) \tag{10}$$

The computations are implemented with JAX and parallelize-able, making it possible to fit the scaling model efficiently, by parallelizing over random initialization trials.

## D.2. Fitting NQS to Scaling Data

Given a scaling dataset $\left\{(N_i, B_i, K_i), L_i^{\text{NN}}\right\}_{i=1}^{m}$, the goal of fitting an NQS is to find $\theta$ that minimizes the scaling loss given by (5):

$$\theta^* = \text{argmin}_\theta \frac{1}{|\text{train}|} \sum_{i \in \text{train}} \mathcal{L}(L_\theta^{\text{SM}}(N_i, B_i, K_i), L_i). \tag{11}$$

In our experiments, we took $\mathcal{L}$ to be the Huber loss between the logarithms of its two arguments, as in Hoffmann et al. (2022).

**Data Filtering.** As described in Sec5.1 , the training dataset for NQS is composed of the IsoFlops training dataset ($N, D$ series, holding $C$ constant) and the IsoTokens training dataset ($B, K$ series, holding $N, C$ constant). Not all elements of the IsoTokens training dataset are suitable to be included in the scaling loss. Because we do not have an implementation of $\nabla_\theta(L^{\text{NQS}})$ that incorporates the Layernorm adjustment, we would like to remove training data points that are expected to be significantly affected by this adjustment, namely those at small batches. In our observations, it suffices to remove data points with $(N, B, K)$ satisfying the following: $L^{\text{NN}}(N, B/2, 2K) > L^{\text{NN}}(N, B, K) - 0.05$. We have access to this information because in the IsoTokens dataset, $B$ are spaced logarithmically, where the successive points are doubled in $B$. This is a rule of thumb that has resulted in a good fit on the filtered training dataset.

**Optimization.** Over the filtered portion of the training dataset, we optimized the target loss in Eq. (11) using the Adam optimizatior, over parallelized random initialization trials, using gradients estimated according to Appendix D.1. Details are given below:

- Initialisations: we used 1000 pseudo-random initialisations, spaced as a Latin hypercube over the following range: $p \in [1.05, 2.5], P \in [10, 100], q \in [0.6, 2.5], Q \in [0.05, 20], \sqrt{R} \in [0.1, 10], r \in [0.6, 2.5], \mathcal{E}_{\text{irr}} \in [1.0, 1.5]$. Note that these values are allowed to move outside of these ranges during the optimization. In the implementation, we parametrized $R$ with $\sqrt{R}^2$.

- Optimization: we used the standard Adam optimizer with gradient clipping (gradients clipped to be within $[-1.0, 1.0]$). Each optimization trial lasts for 5000 iterations.

- Decision: we picked the lowest loss iteration for each random initialization, and then compared them across the initializations to select the final scaling parameters.

In our experiments and on our hardware, the optimization process takes about less than 5 minutes.

# E. Proofs

## E.1. The NQS Model Family and its Degrees of Freedom

Before the derivation, let us review the assumptions and requirements in section 4.1.

We model LLMs as infinite sequences of real numbers, and express the test loss of LLMs as a quadratic over sequences. Let $w_m^* \in \mathbb{R}$ be an square-summable sequence, $H : \mathbb{R}^{\mathbb{N}} \mapsto \mathbb{R}^{\mathbb{N}}$ a positive-definite linear mapping between sequences[6], and $\mathcal{E}_{\text{irr}} \geq 0$. For $w \in \mathbb{R}^{\mathbb{N}}$, define

$$\mathcal{Q}(w) = \mathcal{E}_{\text{irr}} + \tfrac{1}{2}\langle w - w^*, Hw - Hw^* \rangle. \tag{12}$$

We model LLM training as stochastic gradient descent along an finite-dimensional subspace. Let $v_n$ be an orthonormal basis of $H$'s eigenvectors, in non-increasing order of the eigenvalues $\lambda_n$. Let $\gamma, R > 0, w^{(0)} \in \mathbb{R}^{\mathbb{N}}, \xi_n^{(k)} \in \mathbb{R}$ be random, and $\mathbb{W}_N = \text{span}\{v_n\}_{n=1}^N$ for $N > 0$. Define the update:

$$w^{(k)} = w^{(k-1)} - \gamma \, \text{Proj}_{\mathbb{W}_N} \left( Hw^{(k-1)} - Hw^* \right) + \gamma \sum_{n=1}^N \xi_n^{(k)} v_n. \tag{13}$$

We model this with the following assumptions. Let $p > 1, P, q, Q, r, R > 0$.

1. $\mathbb{E}[\lambda_n \times \left( \langle v_n, w^{(0)} - w^* \rangle \right)^2] = P/n^p$,

2. $\lambda_n = Q/n^q$,

3. and $\xi_n^{(k)} \sim \mathcal{N}(0, \sqrt{(R/n^r) \times (1/B)})$ independently.

We want to show that $\mathbb{E}[\mathcal{Q}(w^{(K)})] =$

$$\mathcal{E}_{\text{irr}} + \underbrace{\sum_{n=N+1}^{\infty} \frac{P}{n^p}}_{\mathcal{E}_{\text{app}}(N)} + \underbrace{\sum_{n=1}^{N} \frac{P}{n^p} \left( 1 - \frac{\gamma Q}{n^q} \right)^{2K}}_{\mathcal{E}_{\text{bias}}(N,K)} + \underbrace{\sum_{n=1}^{N} \sum_{k=1}^{K} \frac{\gamma^2 RQ}{Bn^{2q}} \left( 1 - \frac{\gamma Q}{n^q} \right)^{2K-2k}}_{\mathcal{E}_{\text{var}}(N,K,B)} \tag{14}$$

---

[6]Technically, we also assume that $H$ is compact and self-adjoint, to invoke the spectral theorem.

which is the expression we use for the NQS model family. We would also show that the NQS model family, defined as $L^{\mathrm{NQS}}(N, B, K) = \mathbb{E}[\mathcal{Q}(w^{(K)})]$, has at most 7 degrees of freedom.

**Proof.** The update rule gives

$$w^{(k)} - w^{(k-1)} = -\gamma \operatorname{Proj}_{\mathbb{W}_N} \left( H(w^{(k-1)} - w^*) \right) + \gamma \sum_{n=1}^{N} \xi_n^{(k)} v_n. \tag{15}$$

$$= -\gamma \operatorname{Proj}_{\mathbb{W}_N} \left( H \sum_{n=1}^{\infty} \left\langle (w^{(k-1)} - w^*),\, v_n \right\rangle v_n \right) + \gamma \sum_{n=1}^{N} \xi_n^{(k)} v_n. \tag{16}$$

$$= -\gamma \operatorname{Proj}_{\mathbb{W}_N} \left( \sum_{n=1}^{\infty} \left\langle (w^{(k-1)} - w^*),\, v_n \right\rangle \lambda_n v_n \right) + \gamma \sum_{n=1}^{N} \xi_n^{(k)} v_n. \tag{17}$$

$$= -\gamma \sum_{n=1}^{N} \left\langle (w^{(k-1)} - w^*),\, v_n \right\rangle \lambda_n v_n + \gamma \sum_{n=1}^{N} \xi_n^{(k)} v_n. \tag{18}$$

For each $n \le N$,

$$\left\langle w^{(k)} - w^{(k-1)},\, v_n \right\rangle = -\gamma \left\langle (w^{(k-1)} - w^*),\, v_n \right\rangle \lambda_n + \gamma \xi_n^{(k)} \tag{19}$$

$$\left\langle w^{(k)} - w^*,\, v_n \right\rangle = \left\langle w^{(k)} - w^{(k-1)},\, v_n \right\rangle + \left\langle w^{(k-1)} - w^*,\, v_n \right\rangle = (1 - \gamma \lambda_n) \left\langle (w^{(k-1)} - w^*),\, v_n \right\rangle + \gamma \xi_n^{(k)}. \tag{20}$$

Thus $\mathbb{E}\left[ \left( \left\langle w^{(k)} - w^*,\, v_n \right\rangle \right)^2 \right]$

$$= (1 - \gamma \lambda_n)^2 \mathbb{E}\left[ \left( \left\langle (w^{(k-1)} - w^*),\, v_n \right\rangle \right)^2 \right] + \gamma^2 \mathbb{E}\left[ (\xi_n^{(k)})^2 \right] \tag{21}$$

Apply recursively, we get $\mathbb{E}\left[ \left( \left\langle w^{(k)} - w^*,\, v_n \right\rangle \right)^2 \right]$

$$= (1 - \gamma \lambda_n)^{2k} \mathbb{E}\left[ \left( \left\langle (w^{(0)} - w^*),\, v_n \right\rangle \right)^2 \right] + \sum_{j=1}^{k} (1 - \gamma \lambda_n)^{2(k-j)} \gamma^2 \mathbb{E}\left[ (\xi_n^{(j)})^2 \right] \tag{22}$$

$$= (1 - \gamma \lambda_n)^{2k} \frac{1}{\lambda_n} \frac{P}{n^p} + \gamma^2 \sum_{j=1}^{k} (1 - \gamma \lambda_n)^{2(k-j)} \frac{R}{n^r} \frac{1}{B} \tag{23}$$

We also know $w^{(k)} - w^{(0)} \in \operatorname{span}\{v_1, ... v_N\}$, so $\left\langle w^{(k)} - w^{(0)},\, v_n \right\rangle = 0$ for any $n > N$.

$$\mathbb{E}\left[ \left\langle w^{(k)} - w^*,\, H(w^{(k)} - w^*) \right\rangle \right] \tag{24}$$

$$= \mathbb{E}\left[ \sum_{n=1}^{N} \lambda_n \left\langle w^{(k)} - w^{(0)},\, v_n \right\rangle^2 + \sum_{n=1}^{N} \lambda_n 2 \left\langle w^{(k)} - w^{(0)},\, v_n \right\rangle \left\langle w^{(0)} - w^*,\, v_n \right\rangle + \sum_{n=1}^{\infty} \lambda_n \left\langle w^{(0)} - w^*,\, v_n \right\rangle^2 \right] \tag{25}$$

$$= \sum_{n=1}^{N} \lambda_n \mathbb{E}\left[ \left\langle w^{(k)} - w^{(0)},\, v_n \right\rangle^2 \right] + \sum_{n=N+1}^{\infty} \mathbb{E}\left[ \lambda_n (\left\langle w^{(0)} - w^*,\, v_n \right\rangle)^2 \right] \tag{26}$$

$$= \sum_{n=1}^{N} \lambda_n (1 - \gamma \lambda_n)^{2k} \frac{1}{\lambda_n} \frac{P}{n^p} + \sum_{n=1}^{N} \lambda_n \gamma^2 \sum_{j=1}^{k} (1 - \gamma \lambda_n)^{2(k-j)} \frac{R}{n^r} \frac{1}{B} + \sum_{n=N+1}^{\infty} \frac{P}{n^p}. \tag{27}$$

Therefore $\mathbb{E}[\mathcal{Q}(w^{(K)})] = \mathcal{E}_{\mathrm{irr}} + \frac{1}{2}\mathbb{E}\left[ \left\langle w^{(K)} - w^*,\, H(w^{(K)} - w^*) \right\rangle \right]$

$$= \mathcal{E}_{\mathrm{irr}} + \frac{1}{2} \sum_{n=1}^{N} (1 - \gamma \lambda_n)^{2K} \frac{P}{n^p} + \frac{1}{2} \sum_{n=1}^{N} \lambda_n \frac{R}{n^r} \gamma^2 \sum_{k=1}^{K} (1 - \gamma \lambda_n)^{2(K-k)} \frac{1}{B} + \frac{1}{2} \sum_{n=N+1}^{\infty} \frac{P}{n^p} \tag{28}$$

$$= \mathcal{E}_{\mathrm{irr}} + \frac{1}{2} \sum_{n=1}^{N} (1 - \gamma \frac{Q}{n^q})^{2K} \frac{P}{n^p} + \frac{1}{2} \sum_{n=1}^{N} \frac{Q}{n^q} \frac{R}{n^r} \frac{1}{B} \gamma^2 \sum_{k=1}^{K} (1 - \gamma \frac{Q}{n^q})^{2(K-k)} + \frac{1}{2} \sum_{n=N+1}^{\infty} \frac{P}{n^p}. \tag{29}$$

By re-parameterizing $Q =: \gamma Q, R =: \gamma R/2, P =: P/2$, we get:

$$\mathbb{E}[\mathcal{Q}(w^{(K)})] \tag{30}$$

$$= \mathcal{E}_{\text{irr}} + \sum_{n=1}^{N} (1 - \frac{Q}{n^q})^{2K} \frac{P}{n^p} + \sum_{n=1}^{N} \frac{QR}{n^{q+r}} \frac{1}{B} \sum_{k=1}^{K} (1 - \frac{Q}{n^q})^{2(K-k)} + \sum_{n=N+1}^{\infty} \frac{P}{n^p}. \tag{31}$$

Other than $N, B, K$, this function has 7 input arguments: $P, p, Q, q, R, r$ and $\mathcal{E}_{\text{irr}}$. Thus, the model class $L^{\text{NQS}}(N, B, K) = \mathbb{E}[\mathcal{Q}(w^{(K)})]$ has at most 7 degrees of freedom.

**End of proof.**

### E.2. Weight Norm and Learning Rate Scheduling

**Learning Rate Schedule**

The expression for the NQS with a learning rate schedule, where $\gamma_k$ is the learning rate at step $k$, is as follows:

$$L_\theta^{\text{NQS}}(N, B, K, \{\gamma_k\}) = \mathcal{E}_{\text{irr}} + \underbrace{\sum_{n=N+1}^{\infty} \frac{P}{n^p}}_{\mathcal{E}_{\text{app}}(N)} + \underbrace{\sum_{n=1}^{N} \frac{P}{n^p} \prod_{k=1}^{K} \left(1 - \frac{\gamma_k Q}{n^q}\right)^2}_{\mathcal{E}_{\text{bias}}(N,K)} + \underbrace{\sum_{n=1}^{N} \sum_{k=1}^{K} \frac{\gamma_k^2 QR}{Bn^{q+r}} \prod_{j=k+1}^{K} \left(1 - \frac{\gamma_j Q}{n^q}\right)^2}_{\mathcal{E}_{\text{var}}(N,B,K)}, \tag{32}$$

**Weight Norm**

The expected value of the squared weight norm of the NQS in Eq. (6) can be expressed as:

$$||w^{(K)}||^2 = 2 \sum_{n=1}^{N} \left(1 - (1 - \frac{Q}{n^q})^K\right)^2 \frac{P/Q}{n^{p-q}} + 2 \sum_{n=1}^{N} \frac{R}{n^r} \frac{1}{B} \sum_{k=1}^{K} (1 - \frac{Q}{n^q})^{2(K-k)} + s \tag{33}$$

where $s = \mathbb{E}\left[||w^{(0)}||^2\right]$ (Assumption 4.4).

**Proof.**

First, we provide an expression for the term $\mathbb{E}\left[||w^{(K)} - w^{(*)}||^2\right]$.

$$\mathbb{E}[||w^{(K)} - w^{(*)}||^2] = 2 \sum_{n=1}^{N} (1 - \frac{Q}{n^q})^{2K} \frac{P/Q}{n^{p-q}} + 2 \sum_{n=1}^{N} \frac{R}{n^r} \frac{1}{B} \sum_{k=1}^{K} (1 - \frac{Q}{n^q})^{2(K-k)}. \tag{34}$$

The derivation of this equation is analogous to the bias and variance term in App. E.2: $||w^{(0)} - w^{(K)}||^2$ is also a quadratic function in $w$, but because the Hessian, in the place of $\frac{1}{2}H$, is actually the identity in this quadratic, we replace the spectrum $\frac{1}{2} \frac{Q}{n^q}$ in the summation over $N$ with the constant 1.0.

Next, we evaluate $\mathbb{E}\left[||w^{(K)} - w^{(0)}||^2\right]$.

Denote $w_n = \langle w, v_n \rangle$. In the $n$-th eigen-dimension, we have: $w_n^{(K)} - w_n^* = F_{\text{bias}}(w^{(0)} - w^*) + F_{\text{var}}$ where $F_{\text{bias}} = (1 - \frac{Q}{n^q})^K$ and $F_{\text{var}} = \sum_{k=1}^{K} (1 - \frac{Q}{n^q})^{2(K-k)} \xi_k$ is a weighted average of indept. $\xi_k$'s.

This gives us $\mathbb{E}\left[(w_n^{(K)} - w_n^*)(w_n^* - w_n^{(0)})\right]$

$$= \mathbb{E}\left[(F_{\text{bias}}(w^{(0)} - w^*) + F_{\text{var}})(w_n^* - w_n^{(0)})\right] \tag{35}$$

$$= \mathbb{E}\left[-F_{\text{bias}}(w^{(0)} - w^*)^2\right] + \mathbb{E}\left[F_{\text{var}}(w^{(0)} - w^*)\right] \tag{36}$$

$$= \mathbb{E}\left[-F_{\text{bias}}(w^{(0)} - w^*)^2\right] + \mathbb{E}[F_{\text{var}}] \mathbb{E}\left[(w^{(0)} - w^*)\right] \tag{37}$$

$$= \mathbb{E}\left[-F_{\text{bias}}(w^{(0)} - w^*)^2\right] + 0 \times \mathbb{E}\left[(w^{(0)} - w^*)\right] \tag{38}$$

$$= -F_{\text{bias}} \mathbb{E}\left[(w^{(0)} - w^*)^2\right]. \tag{39}$$

The second last line makes use of the zero expectation and independence of $\xi_k$'s.

Therefore $\mathbb{E}\left[(w_n^{(K)} - w_n^{(0)})^2\right]$

$$= \mathbb{E}\left[((w_n^{(K)} - w_n^*) + (w_n^* - w_n^{(0)}))^2\right] \tag{40}$$

$$= \mathbb{E}\left[(w_n^{(K)} - w_n^*)^2\right] + \mathbb{E}\left[2(w_n^{(K)} - w_n^*)(w_n^* - w_n^{(0)})\right] + \mathbb{E}\left[(w_n^* - w_n^{(0)})^2\right] \tag{41}$$

$$= \mathbb{E}\left[(w_n^{(K)} - w_n^*)^2\right] + (1 - 2F_{\text{bias}})\mathbb{E}\left[(w_n^* - w_n^{(0)})^2\right]. \tag{42}$$

And we arrive at the full expression of (6) using the independence of $w_0$:

$$\mathbb{E}[(w_n^{(k)})^2] = \mathbb{E}[((w_n^{(k)} - w_n^{(0)}) + w_n^{(0)})^2] \tag{43}$$

$$= \mathbb{E}[(w_n^{(k)} - w_n^{(0)})^2] + \mathbb{E}[(w_n^{(0)})^2] \tag{44}$$

$$= \mathbb{E}[(w_n^{(k)} - w_n^{(*)})^2] + (1 - 2F_{\text{bias}})\mathbb{E}\left[(w_n^* - w_n^{(0)})^2\right] + \mathbb{E}[(w^{(0)})^2], \tag{45}$$

where due to Assumption 4.4, $\mathbb{E}\left[(w_n^* - w_n^{(0)})^2\right] = \frac{2P/Q}{n^{p-q}}$ (the multiplier 2 is due to the re-parametrization: $P =: \frac{P}{2}$).

Summing over the eigen directions gives us $\mathbb{E}\left[||w^{(k)}||^2\right] =$

$$= \mathbb{E}[||w^{(K)} - w^{(*)}||^2] + 2\sum_{n=1}^{N}\left(1 - 2(1 - \frac{Q}{n^q})^K\right)\frac{P/Q}{n^{p-q}} + s, \tag{46}$$

where $\mathbb{E}[||w^{(k)} - w^{(*)}||^2]$ is already evaluated at the beginning of this proof.

**End of Proof.**

### E.3. Asymptotic Upper Bound for the NQS's Bias Term

In this section we show that symptotically, $(L^{\text{NQS}}(N, B, K) - \mathcal{E}_{\text{var}})$ behaves like Chinchilla.

Specifically, we prove that $\mathcal{E}_{\text{bias}}(N, K) = \frac{1}{2}\sum_{n=1}^{N}(1 - \gamma\frac{Q}{n^q})^{2K}\frac{P}{n^p}$ is $\mathcal{O}(K^{-(p/q-1/q)})$.

It is straightforward to see that $\mathcal{E}_{\text{appx}}(N) = \frac{1}{2}\sum_{n=N+1}^{\infty}\frac{P}{n^p} \sim \mathcal{O}(N^{-(p-1)})$.

**Proof.**

$$\mathcal{E}_{\text{bias}}(N, K) = \frac{1}{2}\sum_{n=1}^{N}(1 - \gamma\frac{Q}{n^q})^{2K}\frac{P}{n^p} \tag{47}$$

$$\leq \frac{P}{2}\sum_{n=1}^{N}n^{-p}\prod_{k=1}^{K}\exp(-\gamma Qn^{-q})^2 \tag{48}$$

$$= \frac{P}{2}\sum_{n=1}^{N}n^{-p}\exp(-2K\gamma Qn^{-q}) \tag{49}$$

We next bound the summation with integrals. To do that, we need to find the regions where the summand is monotone. Take the derivative of the summand $f(n) = n^{-p}\exp(-2K\gamma Qn^{-q})$:

$$\frac{d}{dn}f(n) = (-p)n^{-p-1}\exp(-2K\gamma Qn^{-q}) + n^{-p}\exp(-2K\gamma Qn^{-q})(-2K\gamma Q)(-q)n^{-q-1} \tag{50}$$

$$= pn^{-p-1}\exp(-2K\gamma Qn^{-q})\left(\frac{2q\gamma Q}{p}\frac{K}{n^q} - 1\right) \tag{51}$$

Define $h(K) = (\frac{2q\gamma QK}{p})^{1/q}$. The summand is non-decreasing in $n$ for $1 \le n \le h(K)$, and non-increasing for $h(K) \le n \le N$. Using this monotonicity:

$$\mathcal{E}_{\text{bias}}(N, K) = \frac{P}{2} \sum_{n=1}^{\lfloor h(K) \rfloor} f(n) + \sum_{\lceil h(K) \rceil}^{N} f(n) \tag{52}$$

$$\le \frac{P}{2} \int_{n=1}^{\lfloor h(K) \rfloor + 1} f(n)dn + \int_{\lceil h(K) \rceil - 1}^{N} f(n)dn \tag{53}$$

$$\le \frac{P}{2} \int_{n=1}^{\lfloor h(K) \rfloor} f(n)dn + 2f(h(K)) + \int_{\lceil h(K) \rceil}^{N} f(n)dn \tag{54}$$

$$\le \frac{P}{2} \left( \int_{1.5}^{\lfloor h(K) \rfloor + 0.5} f(n)dn + 2f(h(K)) + \int_{\lceil h(K) \rceil - 0.5}^{N - 0.5} f(n)dn \right) \tag{55}$$

Simplify the integral

$$\int_{x_1}^{x_2} f(x)dx = \int_{x_1}^{x_2} x^{-p} \exp(-cKx^{-q})dx \tag{56}$$

$$= \int_{t_1 = cKx_1^{-q}}^{t_2 = cKx_2^{-q}} (cK/t)^{-p/q} \exp(-t) \frac{d(cK/t)^{1/q}}{dt}dt \tag{57}$$

$$= \int_{t_1 = cKx_1^{-q}}^{t_2 = cKx_1^{-q}} (cK/t)^{-p/q} \exp(-t)(cK)^{1/q}(-1/q)t^{-1/q-1}dt \tag{58}$$

$$= (1/q)(cK)^{-(p/q-1/q)} \int_{t_2 = cKx_2^{-q}}^{t_1 = cKx_1^{-q}} \exp(-t)t^{p/q-1/q-1}dt \tag{59}$$

Define $G(s, (t_1, t_2)) = \int_{t_1}^{t_2} t^{s-1} \exp(-t)dt$ and $c = 2\gamma Q$.

Then we have

$$\mathcal{E}_{\text{bias}}(N, K) \le \frac{P}{2} \frac{1}{b}(cK)^{-(p/q-1/q)} \Bigg( \tag{60}$$

$$G(p/q - 1/q, \ (cK(\lfloor h(K) \rfloor + 0.5)^{-q}, \ cK(1.5)^{-q})) + \tag{61}$$

$$+ 2f(h(K)) + \tag{62}$$

$$G(p/q - 1/q, \ (cK(N - 0.5)^{-q}, \ cK(\lceil h(K) \rceil - 0.5)^{-q})) \Bigg) \tag{63}$$

for convenience, if $y$ is an integer, define $\lfloor y \rfloor = y$ and $\lceil y \rceil = y + 1$, so that we always have $\lfloor y \rfloor + 0.5 = \lceil y \rceil - 0.5$.

Then we get $\frac{\mathcal{E}_{\text{bias}}(N,K)}{\frac{P}{2}(cK)^{-(p/q-1/q)}} \le 2f(h(K)) +$

$$G(p/q - 1/q, \ (cK(N - 0.5)^{-q}, \ cK(1.5)^{-q})) \tag{64}$$

$$\le 2f(h(K)) + G\Big(p/q - 1/q, \ (0, \ \infty)\Big) \tag{65}$$

$$\le 2f(h(K)) + \Gamma(p/q - 1/q) \tag{66}$$

$$\mathcal{E}_{\text{bias}}(N, K) \le \frac{P}{2} (\frac{1}{2\gamma Q})^{p/q-1/q} \Big( 2f(h(K)) + \Gamma(p/q - 1/q) \Big) K^{-(p/q-1/q)} \tag{67}$$

$$f(h(K)) \propto K^{-p/q} \to 0 \text{ as } K \to \infty. \tag{68}$$

We can find sufficiently large $M_1$ such that for all $K > M_1$, $f(h(K)) \le$ e.g. $\Gamma(p/q - 1/q)$ (or any other constant). Therefore $\mathcal{E}_{\text{bias}}(N, K)$ is $\mathcal{O}(K^{-(p/q-1/q)})$. (Holds for any $N$ sufficiently large.)

**End of Proof.**

# F. Experiment Details

## F.1. LLM Model family

We define a model family as a function that maps a requested model size to a fully specified trainable model architecture.

Our main experiments use LLMs trained with the GPT-NeoX suite in the Huggingface Transformers library (Wolf et al., 2020). In our experiments, the requested model sizes are of the form $1e6 \times 2^j$ for integers $j$, ranging from 0.25 to 5120 million parameters. Due to the constraints of the model family, the actual achievable model sizes are not identical to the requested model size. Some of the constraints are: (1) for transformer models, the number of layers and hidden size are required to be integers, and the latter often multiples of 16; (2) we request a certain power law relationship between the number of layers, hidden size and the model size. In short, given a requested model size, we search for an LLM that is close to the requested size, and satisfies the constraints. Details are given below.

To construct the model family, we first fit a power law relationship on the existing Pythia suite of models (Biderman et al., 2023), by running regressing the hidden size ($H$) and the number of layers ($L$) against the model size ($N$):

$$\log(H) \sim p_H \log(N) + a_H, \text{ and } \log(L) \sim p_L \log(N) + a_L.$$

In the pythia family, the intermediate size is always four times the hidden size, and we follow that convention in our model family. We also define the number of heads to be hidden size/16. In Pythia the divisor is $\geq 64$. We chose 16 for convenience, so that we can have an integer number of heads as long as the hidden size is divisible by 16, and be able to construct smaller LLMs that closely match requested model sizes.

Given a requested model size $N_{\text{request}}$, we search in a neighborhood of $N_{\text{request}}$ (10% to 150%), for a value $N'$ that minimizes the difference:

$$\left| N_{\text{NeoGPT}}\Big( H = 16 \times \text{int}(\exp(p_H \log N' + a_H)/16), L = \text{int} \exp(p_L \log N' + a_L) \Big) - N_{\text{requested}} \right|.$$

Here $N_{\text{NeoGPT}}(H, L)$ denotes the count of trainable parameters of a GPT-NeoX LLM constructed with the given hidden size $H$ and number of layers $L$. Said constructed model is the output of the model family mapping for input $N = N_{\text{NeoGPT}}(H, L)$. Where possible, we prefer to use $N = N_{\text{NeoGPT}}(H, L)$ over $N_{\text{request}}$.

## F.2. Datasets

**IsoFLOPs Dataset**. The IsoFLOPs dataset consists of 9 levels, each level contains LLMs trained with a fixed FLOP budget $C$, but with various $N/D$ allocation (by default, we use the Powerlines critical batch size to allocate $D$ to $B, K$). The FLOP budget quadruples between levels, resulting in an overall compute gap of $\times 4^8$. The first 4 levels are used for training (included in the computation of $\mathcal{L}_S$).

**IsoTokens Dataset (B/K variations)**. The IsoTokens dataset is obtained by training LLMs at a fixed model size (10 mil and 100 mil), and consists of up to 5 levels of data points, each level containing LLMs trained at a fixed number of tokens (fixed $D$, varying $B, K$). Between levels, $D$ quadruples, resulting in a $\times 4^5$ gap between the lowest and the highest levels. The first 4 levels are used for training, and the last 2 levels with the highest token counts are reserved for testing.

