# OpenReview forum: "Predicting Large Model Test Losses with a Noisy Quadratic System"
_ICML.cc/2026/Conference — ICML 2026 regular_

### Official Review · Reviewer_qdbP · 2026-03-09

**Soundness:** 3
**Presentation:** 3
**Significance:** 3
**Originality:** 3
**Overall Recommendation:** 5
**Confidence:** 1

**Summary:**

The Authors provide a predictive model that estimates the
pre-training loss of large models from model size (N), batch size (B) and number
of weight updates (K), stressing that this is the first loss prediction model that
can handle changing batch size.

**Compliance With Llm Reviewing Policy:**

Affirmed.

**Final Justification:**

I had no question as the paper was (and remains) much too far from my expertise.
I do not change my evaluation, as my score was already high (5). I explain why
below.

I am new to the ICML Review system. As such, in the first round, I misinterpreted
the advice for Overall Recommendation: I thought that all scores below 4 were to
be used sparingly, whereas the advice was only for scores 3 and 4. As a
consequence of my misreading, the scores I gave for my reviews were 5 and 6 (by
contrast, as an Author, I was generously given 3 to the maximum...).

So my overall score~5 is possibly generous. Moreover, my assessment remains an
educated guess as I am very far from my expertise.

**Key Questions For Authors:**

I have no question as the paper is much too far from my expertise.

**Limitations:**

yes

**Strengths And Weaknesses:**

Soundness
3: good

The mathematics seem sound. There are also plenty of numerical experiments, but
that I am not able to judge.


Presentation
3: good

This paper pre-supposes that the reader has enough background about loss
prediction models. This is not my case and, for this reason, I found the paper hard to follow.


Significance
3: good

I think that the paper does address a relevant problem, and can advance
practice in machine learning.


Originality
3: good

This is hard to answer as I do not have enough perspective on the subject.

---

> ### Author Rebuttal · Authors · 2026-03-31
>
> We thank the reviewer for verifying the mathematical derivations, and for the positive assessment.

---

> > ### Author Rebuttal · Reviewer_qdbP · 2026-04-01
> >
> > As I wrote "I have no question as the paper is much too far from my expertise.", I select (a) Fully resolved. My score was already high; I do not change it.

---

### Official Review · Reviewer_51sF · 2026-03-12

**Soundness:** 3
**Presentation:** 3
**Significance:** 3
**Originality:** 2
**Overall Recommendation:** 4
**Confidence:** 2

**Summary:**

This paper focuses on the problem of predicting the pre-training test loss of large language models. Based on the Noisy Quadratic Model (NQM) and linear regression dynamics, this work proposes the Noisy Quadratic System (NQS), a mechanistic predictive model that integrates model size, batch size, and the number of weight updates. The authors also consider the effect of LayerNorm on the effective learning rate to model small-batch training dynamics more accurately. The proposed method is validated against LLM training runs, demonstrating superior performance over existing empirical scaling laws.

**Compliance With Llm Reviewing Policy:**

Affirmed.

**Final Justification:**

The rebuttal addressed my concerns and I maintain my positive score.

**Key Questions For Authors:**

My main question was already posed in the "strengths and weaknesses". Other minor suggestions:

1. **Typos:** line 166, "senarios", line 193, " innner", line 334, "specifed".

2.  While introducing batch size extends the predictive model's capability, it also adds new parameters and potential uncertainty, which may hurt the final prediction performance. I think providing an estimation error bound would help analyze how this added complexity impacts the model's overall performance.

**Limitations:**

Yes.

**Strengths And Weaknesses:**

# Strengths

1. Compared with existing empirical methods that ignores the batch size, this paper successfully incorporates it into the loss prediction framework, which makes the proposed method novel and more practical for real-world scenarios.

2. The proposed model has strong and stable extrapolation performance. To be detailed, it reliably predicts test losses for large scale models.

3. The method can promisingly guide model selection. This is particularly useful for users who need to find the optimal training setup under time and memory constraints.

# Weaknesses

1. While the NQS formula is clear, a formal discussion on the estimation error bound will greatly enrich the theoretical contribution of this paper.

2. The optimization relies on a separate grid search to mitigate the numerical issues for parameter $s$. Unifying this process would make the method more integrated.

---

> ### Author Rebuttal · Authors · 2026-03-31
>
> We thank the reviewer for their careful review and for recognizing the novelty and strong performance of our method. In particular, we appreciate the suggestion to include error bounds to assess the robustness of our methodology. Following this suggestion, we conducted additional experiments to estimate error bounds for the NQSs, which provided support for the reliability of our predictions.
>
> ---
>
> **W1 & Q2:**
>
> **A formal discussion on the estimation error bound will greatly enrich the theoretical contribution of this paper, and providing an estimation error bound would help analyze how the added complexity (from modelling batch size) impacts the model's overall performance.**
>
> Unfortunately, we do not yet have a formal mathematical estimation error bound, as the inference procedure is difficult to analyze (e.g. error is computed on the log scale).
>
> As an alternative, to assess the inherent uncertainties in the model, we produced empirical error bounds on the NQS’s predictions via subsampling: re-fitting the NQS on a random 50% of the training LLM runs; repeat for 100 trials to construct a 90% confidence interval. The bounds gave high confidence in our predictions. The links below provide visualizations of the error bounds; we will include these figures in the final version.
>
> Figure: https://files.catbox.moe/kinhrt.png
> Backup Link: https://i.imgur.com/jOIJaHG.png
>
>
> *Figure 8*: The NQS is robust to changes in the training set. Shown is a 90% confidence interval around the NQS predictions, constructed using 100 trials, each using a random subset (a 50% subsample) of the training runs to infer the NQS parameters.
>
> Figure: https://files.catbox.moe/as7gl0.png
> Backup Link: https://i.imgur.com/FGZf3vv.png
>
> *Figure 9*: The NQS is robust to changes in the training set: the 90% confidence intervals are tight relative to the range of the predictions.
>
> ---
>
> **W2: The optimization relies on a separate grid search to mitigate the numerical issues for parameter. Unifying this process would make the method more integrated.**
>
> We agree that estimating s more efficiently would make the model easier to use, and the numerical issues that prevented this integration appear to be solvable with careful engineering. As mentioned in the discussions, we would like to address this in future works.
>
> ---
>
> **Q1: Typos**
>
> Thank you for pointing out the typos! We will fix them in the final draft.
>
> ---

---

> > ### Author Rebuttal · Reviewer_51sF · 2026-04-03
> >
> > The additional analyses and results fully resolve my concerns. Therefore, I am maintaining my score.

---

### Official Review · Reviewer_8bXD · 2026-03-12

**Soundness:** 3
**Presentation:** 3
**Significance:** 2
**Originality:** 3
**Overall Recommendation:** 4
**Confidence:** 3

**Summary:**

The paper extends the chinchilla scaling law by including batch size used to train in the formulation. This is done by considering a more involved model of NQS which models the variance of parameters as the training progresses.  The authors show this more involved model of loss is able to better model out-of-distribution compute flops, where the test flops are 100x or more than the max train flops used.

**Compliance With Llm Reviewing Policy:**

Affirmed.

**Key Questions For Authors:**

1. In Figure 3, the effective learning rate seems to help only when batch size < 1K. Is large models with small batch size a realistic case in practice we need to account for?
2. Is there any non-intuitive advice widely applicable for practitioners that can be inferred from this model? For e.g., the Chinchilla model proposed scaling parameters and data proportionally which was not the norm before the Chinchilla (Hoffmann et al.) paper.

**Limitations:**

yes

**Strengths And Weaknesses:**

Strengths:
1. The method is theoretically sound.

Weaknesses:
1. The model relies on the assumption that effective learning rate decays as the norm of the parameters. Furthermore there is a very strong functional form for the norm of the parameters as well. Though they are well grounded in the linear theory case, I am not sure how applicable these assumptions and strong functional forms are to larger transformer models (> 10B parameters)
2. The empirical evaluation section is weak. Justification of the model presented would typically require stronger and in general more evaluations.

---

> ### Author Rebuttal · Authors · 2026-03-31
>
> We thank the reviewer for the positive assessment, and for recognizing the theoretical soundness of our approach.
>
> ---
>
> *Response to Weaknesses*
>
>
> **W1: Though grounded in the linear theory case, the effective learning rate decay assumption and the functional form of the weight norm may not apply to larger transformer models (>10B parameters).**
>
> As introduced in section 3.3 of our draft, the decay of effective learning rate in neural network training is supported by theoretical work on common NN optimizers: van Laarhoven, 2017 found the effective learning rate to be inversely proportional to the weight norm. Their analysis applied to the Adam optimizer (which we used to train the LLMs in our dataset) and was empirically assessed on neural networks.
>
> The main assumption we made was that the NQS’s weight norm is a good representation of the LLM weight norm throughout training. To reduce the gap between NQS and LLM, we used a functional form of the initial LLM weight norm that calibrated the NQS dynamics.
>
> As the reviewer pointed out, this functional form of the initial weight norm, \sqrt(N)*0.02, may not apply to every model family, and it would be prudent to test and adjust the formula as needed. In the python package we plan to release, we will include a functionality for the user to adjust this formula (weight norm as a function of the model size) to adapt to their use case.
>
> Regarding the scale of the LLM models, we believe that our largest model (2B parameters) already lies in a regime where key structural properties of LLMs have stabilized. For example, the Pythia family [Biderman et al.], spanning 70M–12B, is designed to study scaling behavior and is considered sufficiently uniform in structure. This gives us some confidence that our conclusions would extend to larger models, but empirical verification of this would be important future work.
>
>
>
> **W2: Justification of the model presented would typically require stronger and in general more evaluations.**
>
> We agree that running experiments on more distinct LLM workloads will provide stronger evidence for generalisation. In the submission, we only presented two distinct workloads (Llama + LM1B, and Pythia + OWT2). In either of the two workloads presented, we did have around 100 distinct LLM runs where on each point we evaluated the NQS prediction against the ground truth.
>
> ---
>
> *Answers to Questions*
>
> **Q1: In Fig. 3, the effective learning rate seems to help only when batch size < 1K. Is large models with small batch size a realistic case in practice we need to account for?**
>
> Thank you for the insightful question. We do not know for sure if the effective learning rate effect would be significant in frontier-scale models trained at very large batch sizes. That said, because frontier-scaled models would be trained for many more iterations, and even small mini-batch noise (from large batches >>1K) can still accumulate into a significant variance term over the many iterations, it is possible that the need to use effective learning rate could still arise.
>
> In general, small batch training is relevant in data-constrained settings, as well as GPU-constrained settings:
>
> - Data Constraint: according to Marek et al., 2025, holding constant the total number of tokens, models trained at smaller batch sizes outperform those trained at larger batch size.
>
> - GPU Constraint: in academic settings, experiments are often limited by the total available GPU memory, where there is a trade-off between maximum batch size and model size.
>
> **Q2: Is there any non-intuitive advice widely applicable for practitioners that can be inferred from this model?**
>
> Thank you for raising this point. The NQS model is consistent with some very recent findings on the selection of batch sizes:
>
> - Bergsma et al., 2025 found a power law relationship between optimal batch size and token count (as opposed to between optimal batch size and total compute).
> - Marek et al., 2025 showed that under a generous time budget, smaller batch sizes are preferable for higher performance.
> - Zhang et al., 2024: “Empirical Takeaways” 1 to 3, namely:
> 1. In Chinchilla settings, the Critical Batch Size (CBS) increases when model size N and data size D are jointly scaled up.
> 2. If we scale up training duration D while keeping N fixed, the CBS increases to a similar degree.
> 3. The CBS remains nearly invariant when scaling up N while keeping D fixed, suggesting that CBS weakly depends on model size N but more strongly depends on data size D.
>
> In the submission, we did not draw attention to these lessons, but such discussions would strengthen the paper, and we are happy to add them to the final draft.

---

> > ### Author Rebuttal · Reviewer_8bXD · 2026-04-03
> >
> > I thank the reviewers for their rebuttal. I still think that there is a very strong functional form and am not very sure about how generalizable these results are.
> > But nonetheless I believe that my score stands justified.

---

### Official Review · Reviewer_81nR · 2026-03-14

**Soundness:** 3
**Presentation:** 4
**Significance:** 4
**Originality:** 3
**Overall Recommendation:** 5
**Confidence:** 2

**Summary:**

In this paper, the authors introduce a mechanistic loss model(NQS) that expresses the test loss as a function of N,B,K. The proposed model is inspired by theoretical models of neural network scaling dynamics. Unlike other models, it has a closed form solution that admits numerical approximation. The paper advocates for loss-prediction as a better alternative to heuristic-based laws. Experiments with different model sizes show that they outperform Chinchilla method 3.

**Compliance With Llm Reviewing Policy:**

Affirmed.

**Final Justification:**

I have no follow up questions as I am not an expert in this area and I keep my high score.

**Key Questions For Authors:**

1. The proposed approach is derived from SGB based model. How does change in the optimizer of different lr schedules affect the analysis?

**Limitations:**

Yes

**Strengths And Weaknesses:**

**Soundness**
1. The paper is largely sound. The experiments are well designed and support the claim.

**Presentation**
1. The paper is well-structured and easy to follow.
2. The experimental section is clear. The background on training dynamics, limitations are clearly described.

**Significance**
1. The paper solves a highly relevant problem in the era of LLM training where the resources are scarce in many scenarios. The proposed solution is specifically beneficial to real-world applications where the size of models and data are huge and the ability to predict loss as a function of N,B,K is valuable. Even from the scientific point of view, the proposed approach can help influence various directions of research.

---

> ### Author Rebuttal · Authors · 2026-03-31
>
> We thank the reviewer for the encouraging feedback. We appreciate their recognition that our work addresses a highly relevant problem in LLM training and has the potential to inform future research.
>
> **Q1: How does change in the optimizer or different lr schedules affect the analysis?**
>
> In our experiments, we found that an SGD-based NQS could model more sophisticated optimizers in the LLM. We believe this is because the NQS roughly approximates the effect of various optimization choices, including SGD with moment and Adam, via a constant quadratic preconditioning of SGD. We do think some optimization choices, like Muon or other optimizers that exploit the matrix structure, cannot be modelled in this way. We would be excited to explore this in future work to extend the NQS.
>
> For lr schedules, we saw that the NQS can fit LLMs trained with a constant lr schedule or a cosine lr schedule, but the inferred NQS parameters are different: for LLMs trained with SGD and constant lr, the irreducible error term was higher and the hessian spectrum was steeper (i.e. less pre-conditioning). We will include this comparison in the appendix for the final draft.

---

> > ### Author Rebuttal · Reviewer_81nR · 2026-04-03
> >
> > Thank you for the answers. I do not have any follow up questions. My score is already high. Will keep it.

---

### Decision · Program_Chairs · 2026-04-30

**Decision:**

Accept (regular)

**Comment:**

This paper introduces the Noisy Quadratic System (NQS), a mechanistic model predicting pre-training loss using model size, batch size, and update count—outperforming Chinchilla, especially under extreme compute extrapolation (up to 1000×). It offers practical value for optimal training configuration under real-world constraints.

Strengths include strong theoretical grounding and robust empirical validation across LLM workloads. Weaknesses involve assumptions about weight norm dynamics and lack of formal error bounds, though authors addressed these via empirical confidence intervals in rebuttal.

Given  clear significance, and solid responses, I recommend acceptance.